# Acceptance of evolution by high school students: Is religion the key factor?

Graciela da Silva Oliveira[1,2], Giuseppe Pellegrini[2,3,4], Leonardo Augusto Luvison Araújo[2,5], Nelio Bizzo[2,5,6]*

1 Departamento de Biologia e Zoologia, Instituto de Biociências, Universidade Federal de Mato Grosso, UFMT, Cuiabá, Brazil, 2 Research Nucleus on Education, Epistemology and Outreach of Evolution "Charles Darwin", USP, São Paulo, Brazil, 3 Dipartimento di Sociologia e Ricerca Sociale, Università degli Studi di Trento, UniTrento, Trento, Italy, 4 Observa–Science in Society, Vicenza, Italy, 5 Faculdade de Educação, Departamento de Metodologia de Ensino e Educação Comparada, Universidade de São Paulo, USP, São Paulo, Brazil, 6 Departamento de Ciências Exatas e da Terra, Instituto de Ciências Ambientais, Químicas e Farmacêuticas, Universidade Federal de São Paulo, UNIFESP, Diadema, Brazil

* bizzo@usp.br, bizzo@unifesp.br

## Abstract

The idea of biological evolution is not accepted by many people around the world, with a large disparity amongst countries. Some factors may act as obstacles to the acceptance of evolution, such as religion, a lack of openness to experience, and not understanding the nature of science. Although the strength of the association between evolution acceptance and non-scientific factors varies among studies, it is often assumed that resistance to evolution is the byproduct of a religious background. Some studies are even more specific and try to associate the acceptance of evolution with precise religious affiliations. We aimed to explore the strength of associations among nationality, religion, and the acceptance of evolution by students using multiple correspondence analysis (MCA) and statistical tools, with nationwide samples from two different countries. Here, we show that wider sociocultural factors predict the acceptance of evolution to a higher degree than a religious background. We carried out two nationwide data collections that allowed us to compare differences in the acceptance of evolution in Italy and Brazil by high school students who declare to belong to the same religion in the two countries. Roman Catholic students showed significant differences between the two countries, and the gap between them was wider than between Catholics and non-Catholic Christians within Brazil. Our conclusions support those who argue that religious affiliation is not the main factor in predicting the level of evolution acceptance. The sociocultural environment and the level of evolutionary knowledge seem to be more important in this regard. These results open up new interpretative perspectives and provide a better understanding of attitudes towards evolution.

## Introduction

The acceptance of evolution has been a central topic in evolution education in at least the last three decades [1,2]. Researchers have examined numerous factors that may act as obstacles to

**Data Availability Statement:** Data are held in a public repository in the following url: https://github.com/easouza85/Nelio-Bizzo-Project.

**Funding:** GP: OBSERVA Science in Society (Vicenza/Italy – www.observa.it) NB: MEC (CAPES/

PAEP 6726/2012-76, Brasília, Brazil; https://www.
gov.br/capes/pt-br), MCTI/CNPq (308899/2011-3,
308877/2015, 440680/2019-0, and 309073/2020-
0, Brasília, Brazil, CNPq - Conselho Nacional de
Desenvolvimento Científico e Tecnológico —
Brasília, Brazil; www.gov.br), FAPESP (proc. 2016/
05843-4, São Paulo, Brazil; www.fapesp.br) and
NAP EDEVO-Darwin (Pro-Reitoria de Pesquisa,
Universidade de São Paulo – São Paulo, Brazil;
http://prp.usp.br/nucleos-de-apoio-a-pesquisa-
naps/); LALA: FAPESP (2020/07961-0 – São Paulo,
Brazil; www.fapesp.br).

**Competing interests:** The authors have declared
that no competing interests exist.

the acceptance of evolution, such as religiosity, knowledge of evolutionary theory, the understanding of the nature of science, scientific aptitudes, and psychological conflicts, among others [3–6]. A major association seems to exist among three main factors: the understanding of key biological concepts, religious factors, and the understanding of the nature of science. However, the data require further investigation.

A systematic literature review of the current state of research regarding students' and teachers' acceptance of evolution across Europe found that the level of acceptance of evolution in different educational settings is ambiguous. The authors argue that similar samples and a standardized assessment of the acceptance of evolution are necessary for cross-country comparisons [7].

The use of different instruments to measure evolution acceptance could be a cause of conflicting research results. A variety of tests have been proposed, including the Measure of Acceptance of the Theory of Evolution (MATE), which has been widely used for more than 20 years and recently revised [8], the Inventory of Student Evolution Acceptance (I-SEA) [9], the Generalized Acceptance of EvolutioN Evaluation (GAENE) [10], etc. It should be considered that the studies cited above differ in their contexts of comparison in relation to ages, countries, and socioeconomic levels, as well as the number of data collected and measurement statistics. However, inconsistencies in the results were found even in the analysis of the same group with different instruments measuring the acceptance of evolution.

Barnes et al. [6] conducted separate analyses using six different evolution acceptance instruments with the same group of students, a large sample of around 2,300 university students. The instruments were administered in a random order to avoid bias due to differential attention and response readiness. The results were surprising and showed that different instruments led to different results for the measure of evolution acceptance, including conflicting ones. These results may also be interpreted as an indication that some instruments need methodological revisions in the search for possible distortions and biases.

Additionally, the research findings revealed a complex relationship between understanding evolutionary concepts (e.g. natural selection) and evolution acceptance [10–13]. The same lack of consensus exists when measuring the level of evolution knowledge, given that several alternative assessment tools are available and may lead to different results [14,15]. A study with secondary school students (aged 14–16) in the United Kingdom found that teaching genetics before teaching evolution had a significant impact on improving evolution understanding but did not result in a significantly increased acceptance of evolution [16]. This reflects a weak correlation between the knowledge and acceptance of evolution, which also appears to be present in undergraduate students. In Chilean students of 15–16 years old, it was found that including instruction on the nature of science in the class on evolution improved their acceptance significantly [17].

Kuschmierz et al. [7] identified 56 papers from the period of 2010–2020 regarding students' and teachers' knowledge and acceptance of evolution across 29 European countries. However, according to the authors, the research findings were hard to compare due to the application of different instruments and the assessment of different key concepts. The available database was not sufficient to obtain reasonably comparable data from European countries. In addition, the ambiguous results of the research demonstrate multiple challenges regarding measurement of evolution education, such as (a) inadequate definitions of key constructs (attitudes, acceptance, knowledge, and understanding are often not defined or inconsistently used in publications); (b) the application of diverse measurement instruments (different evolution acceptance instruments may produce different results, even when applied to the same population); (c) the multidimensionality of knowledge about evolution (most instruments focus on single evolutionary constructs, such as natural selection and related concepts).

The authors [18] then developed an instrument to measure attitudes and understanding across Europe and beyond, called the "Evolution Education Questionnaire on Acceptance and Knowledge" (EEQ). The measurement instrument was translated into several European languages, in the various Romance, Germanic, and Slavic branches, and was recently applied to 9200 first-year university students in 26 European countries [19].

This is a recent initiative, and thus, due to a lack of standardized assessment procedures in the existing literature, previous results should be used with caution when trying to compare countries, as there are several limitations, such as sampling biases, etc. [18].

Religious beliefs are among the factors most investigated as predictors to the low acceptance of evolution in several studies, especially among some age groups, religious affiliations, and nationalities, mainly Christian American students [10,20,21]. Barnes et al. [22] conducted a study with 1898 students in eight states of the United States and found that 56.5% of students perceived that evolution is atheistic, with a higher percentage of Catholic and other Christian students. This perception among students may lead to an apparent conflict between their personal religious beliefs and evolution [23].

In addition to Christians, those of the Muslim faith have been investigated, although the debate between several religions and evolution is more typically addressed in Western contexts [24]. However, multifactorial models have found that once religiosity and other measures are accounted for, the amount of variance in acceptance of evolution is greatly reduced [13,25]. Mantelas and Mavrikaki [26] examined the acceptance of evolution and its relationship with religiosity among Greek university biology students. A rather weak correlation was found between the two factors.

Evolution is controversial among both religious and non-religious individuals, and inconsistent views of evolution have been evidenced in both groups [24,27]. The picture appears to not be static, as a sample of U.S. education professionals with a wide range of religious commitments showed a significantly higher level of acceptance than previous studies [28]. Moreover, the level of public acceptance of evolution has increased in the United States in the last decade, and education seems to play a significant role [29]. Thus, in certain social contexts, religiosity may be a main component by itself, while in others, evolution acceptance may depend on other factors.

There is a tendency, which cannot be taken as unanimous, that associates the belief in any God (deism) as intrinsically anti-evolutionist [30]. One can easily remember the several references to the "Creator" in the Origin of Species, especially in its second edition, showing that this direct link is far more complex than it may seem at a first sight. The high heterogeneity of data available and the well-known methodological issues around gathering sensitive opinions and beliefs do not allow for a narrow focus on the search for one isolated variable. However, it has been often assumed that resistance to evolution is somehow a byproduct of religious background, especially in the United States [22,31].

Some researchers have focused closely on specific religious groups, such as Pentecostal Christians, known for their literal reading of sacred texts and strong opposition to evolution [32]. One study was carried out in Brazil, where these religious groups have a growing influence, and tried to establish a direct correlation between the frequency of Pentecostal Christians and the level of rejection of evolution between the students of two high schools [33]. A written instrument was created, based on a well-known one [8], and 10 students were invited for semi-structured interviews [33]. Although no direct correlation was found, the author concluded that religious beliefs are important elements that shape students' ideas, possibly introducing "constraints that might hinder the understanding of evolutionary theory" [33, p. 63].

Another piece of research developed a survey instrument with a nationwide sample of the British population about the general public's views of evolution. As the study had a closer

focus on religion, there was oversampling of five different religious affiliations: Anglicans (or Episcopalians), Catholics, Muslims, Pentecostal Christians, and Independent Evangelical Christians. The results showed a strong rejection of evolution by Muslims, Pentecostals, and Independent Evangelical Christians, contrary to the position of non-religious people, Anglicans, and Catholics, who showed high agreement to statements about human evolution. The study revealed that the frequency of religious service attendance is a large significant predictor of evolution rejection among most religious groups [24, p.88]. The general conclusion was that religious affiliation and the degree to which individuals participate in religious practices affect the acceptance of evolution as a valid theory.

A recent study with a large sample of almost 8000 undergraduate biology students in different states of the United States searched for correlations between religious affiliation and variables such as the acceptance and understanding of evolution. Using a Likert scale instrument, they found that Muslim undergraduate students showed slightly higher evolution acceptance levels than Protestant students, but significantly lower levels than Catholic, Jewish, Buddhist, and Hindu students [34].

However, some researchers have been questioning the common view that religious affiliation is the main factor to the acceptance of evolution. They argue that it is more likely that people form beliefs congruent with their broad cultural identity, which includes but is not limited to religious belief [35,36]. However, most articles arguing in this direction are case studies and philosophical or legal debates about evolution vs creation in the school curricula, mostly in the U.S. context [37]. In addition, with few exceptions, such as the reports by Miller et al. [2] and Clément [36], most studies have focused on a specific culture or country, even if with large samples.

Our main research question tried to address this complex picture considering the same religious affiliation in two different countries with deep sociocultural differences. Catholic Christians in Italy and Brazil have several similarities, including many family connections owing to immigration history. Brazil is the country with the highest number of Catholic Christians in the world, and Italy is the hub of Roman Catholicism. They follow the same basic regulations from the Vatican, which include explicit views on biological evolution and a non-literal interpretation of sacred texts. However, there are deep sociocultural differences due to many complex factors, including education. A survey comprising 192 Brazilian students was carried out in Brazil in 1990, which found a great deal of Lamarckian views [38]. One should also consider that in the last three decades, there has been a significant rise in the creationist movement in Brazil [39]. One Brazilian state minister declared in 2019 that the Evangelical Church should fight against the introduction of the theory of evolution in public schools, before her recent nomination [40]. Another member of the federal government declared in an open ceremony that Intelligent Design (ID) should be introduced in Brazil's public schools as "a counterpoint to the theory of evolution", assuming that there would be a "creationist theory" to be taught in science lessons starting at the primary level [41]. Research carried out in Brazil had already showed that creationism had a significant influence on young science teachers [42].

Italy is a highly industrialized country within the European Union. Research on the public understanding of science has been frequent, with nationwide surveys since 2004 [43]. However, specific research on the knowledge and acceptance of evolution in Italy was scarce at that time. Whereas Italy is included in broader international surveys, it accounts only for a few targeted case studies that vary in types of audience and goals. Despite the scarce data, researchers indicated that rejection of the evolutionary theory is neither widespread nor deeply rooted in Italian society [44], which was confirmed by Pew Research Center reports. The same source recently confirmed the tendency of the rise of the creationist perspective in Brazil within adults, with a considerably lower acceptance of evolution by Christians in Brazil (51%) than in Italy (74%) [45].

## Methodological issues

Despite the similarities between Italy and Brazil, there is a marked difference between the two sociocultural contexts. However, some methodological issues should be properly addressed to avoid some sources of significant bias in the statistical tools to be used. The sample design was an important step, as nationwide samples best represent the complexity of the several sociocultural influences. As we were targeting the main religious group in both countries (Catholics), no oversampling was necessary. Non-Catholic Christians would be part of the sample, but no specific conclusions could be drawn considering the different religious affiliations within the group.

Brazil is a country where religious practices are diverse and encompass several different belief systems and traditions, which reflect elements of European (Catholic and Protestant), African, Oriental, and indigenous religions, among others. This diversity has its foundation in the colonization process of European immigrants from 1500 onwards, African descendants, and local indigenous tribes [46].

The group of non-Catholic Christians is very heterogeneous, and our database aggregated several different affiliations in the two countries: Orthodox, Lutheran, Presbyterian, Baptist, Adventist, Jehovah's Witnesses, Assembly of God, Christian Congregation of Brazil, International of God's Grace, World of God's Power, and the Universal Church of the Kingdom of God. Therefore, we consider that the comparison between Catholics in the two countries was more suitable for this study, as it is a more homogeneous group. The term "evangelical" represents the universe of non-Catholic Christians considered by Brazilian society as "Brazilian Protestantism", which is constituted by historical Protestants, Pentecostals, and Neo-Pentecostals and led by Pentecostals, with a massive dissemination by Neo-Pentecostals. Brazil is considered the country with the largest number of Roman Catholics in the world. However, this hegemony has been decreasing in recent decades, which reveals a changing trend in the religious composition of the population [47]. Data from the population census in Brazil point to the decline of the Roman Catholic religion and a continuous rise of protestant groups in the last decades [48].

The age level was an essential component for sampling, as elderly people tend to be more conservative than young students. Taking advantage of an international project focusing on youngsters who were beginning high school level in both countries, we prepared separate sets of questions about evolution so that we had the very same questions answered by students of the same age in the same time frame, during the first months of the year 2014. Targeting students at the same age level in two different countries allows for closer comparisons than multi-age-wide groups. In the next section, a more detailed description of the sampling procedures in both countries is given.

There is an important distinction that we must keep in mind when comparing the concepts of belief and acceptance. Differences between them have long been discussed in the philosophical literature and the specific field of evolution education [25,27,43,49]. The overall picture takes belief as a subjective disposition to consider *p*. simply as a premise under the influence of the wide cultural context. A well-known distinction is recognized between belief and conjecture since the latter is not committed to sensible things and applies even to the world of shadows. In this context, it is important to separate the notions of belief of classical philosophers, such as Plato and Aristotle, from the revision carried out by Thomas Aquinas, who considers belief in terms of "thinking with assent" related to faith [49]. In our theoretical framework, belief refers to the sensible world only, with the reality of things that exist. However, one can take *p*. as a belief, but it does not in itself imply an admission of the objective validity of the notion under consideration. Meanwhile, to accept *p*. implies not only recognizing the

existence of some objective reality but also a further step in terms of judgment, admitting *p*. as valid under some specific circumstances; in short: "acceptance implies a commitment to a policy of premising that *p*." [50].

The instrument used in the recent work by Kuschmierz et al. [19], ATEVO, relies on eight items with statements about evolution, and students show their degree of agreement with each of them [51]. In the German context, the acceptance of evolution is defined as a "positive attitude" ("positive Einstellung") towards evolution. In this regard, "attitude" ("Einstellung") is defined as the level of association between a term or fact and its subjective evaluation, in terms of strength ("Starke") and/or ability to bind with other terms or ideas ("Valenz"). On the opposite side, a "negative attitude" ("negative Einstellung") is seen as "rejection" ("Ablehnung"). However, the result of ATEVO test may be the result of very different processes. On the one hand, "it remains unknown whether a person answers negatively for most items about evolution because he/she has a 'bad feeling'" ("schlechtes Gefühl") about evolution, in the sense of a preexisting emotional refusal to think on the subject. On the other hand, the rejection may have been the result of a long reflection leading to a rational conclusion that evolution is not plausible [51, p. 12].

We adopted a clear definition of evolution acceptance despite the complexity of the discussion on the subject in different languages, taking acceptance as the expression of explicit recognition of the objective validity of known scientific statements about evolution under absolute anonymity. This definition considers two steps. The first is associated with scientific statements about evolution, which must be clear and well known, avoiding issues under discussion, for instance, about the origin of life. Students must show not only a positive attitude towards evolution but also express clearly that a statement based on biological evolution is considered a valid premise to construct a judgement about the real world.

The second step refers to objective conditions in which a person may admit his/her positive judgment about a certain scientific statement. One may know a proposition but to refuse to show public recognition of it, or, on the contrary, manifest an opinion different from his/her deep feeling due to the suspicion that it may reach a wider audience. The perception of secrecy in everyday life is actually a major focus of concern worldwide. Strong evidence published in the literature shows that anonymity reduces social desirability distortion and increases self-disclosure [52,53].

Therefore, anonymous participation must be guaranteed to every person who is responding to questions about religion-related subjects, such as biological evolution, especially when surveys are carried out within conservative social contexts. Under secrecy, a person may accept a proposition as a valid premise for argumentation, making inferences, deliberations, etc. as a mental, individual, and innermost act. However, this may not happen if there is a perception of the possibility of third parties identifying the individual opinion. This means, for instance, that phone calls or traceable connections using electronic devices such as smartphones could be perceived by respondents as possible ways of infringing the inviolability of the anonymous character of the individual manifestation.

In our case, the students answered questions in their school environment and were told their answers would be under absolute anonymity. Previous analyses of this data bank showed surprisingly high mean scores of evolution acceptance within fundamentalist religious groups in Italy and Brazil [54]. This was also the case in a previous survey using a different methodology, carried out in 2010–11, where roughly half of Evangelical high school students in Brazil declared under absolute anonymity that their religion was not opposed to evolution [55].

In line with the tantalizing results of the study by Barnes et al. [6] already mentioned, our research team proposed a revision to some methods adopted in the previous survey. As mentioned before, we were carrying out data collection as part of an international project focusing

on students' interests and attitudes on science and technology, which had started in 2007 [56]. The main research instrument was based on Likert scale items but allowed a separated section, which we added in both countries. However, we considered a different approach for these evolution items, as we were going to present students with factual statements that were regarded as right or wrong by scientists.

Items following the Likert attitude data consist of favorable or unfavorable statements about an entity, admitting a spectrum of responses ranging from strong positive to strong negative answers, with a numerical scale from 1 to 5. The resulting data can be analyzed using several models that underlie latent variables. However, attitudes as well as response styles can affect the result. It is well known that numerical comparisons rely on the degree of precision of measures, for instance, to estimate the mean height of a group of people. The significance of the different statistical analyses are dependent on the precision of the instrument used in data collection. Likert scaling is used in instruments measuring attitudes, beliefs, and opinions about statements, allowing for an expression of moderation of opinion from agreement to disagreement. The key is to find ways of calibrating how strongly or mildly a statement should be worded. These scales are therefore suitable for issues related to open constructs and less suitable for scientific claims based on presumed factuality. An item likely to produce extreme responses, either full agreement or strong disagreement, would do a poor job of discriminating across the full spectrum of respondents [57]. Therefore, Likert scales may not be a good choice with statements expressing well-known scientific facts. In addition, in the case of theories of evolution, factuality is supported by the school context in which the questionnaires were filled out. The battery of items on evolution was proposed with dichotomous questions, given that each of them can have minimal variability.

When evolution is considered, opinions can be expressed in terms of favorable and unfavorable subjective statements, such as "There is a significant body of data which supports evolutionary theory" [33], "There is strong, reliable evidence to support the theory of evolution" [24], or "I think there is reliable evidence to support the theory that describes how humans were derived from ancestral primates" [33]. In these three examples, latent variables can be inferred in terms of personal attitudes, which can range from weak to strong, and grading would help to correctly appraise the level of agreement. However, well-known factual scientific statements cannot be graded as weak or strong scientific facts. For instance "The age of the Earth is at least 4 billion years" [33] or "I think that humans and apes share an ancient ancestor" [15], should be recognized as sound scientific statements (or not). We have here a black/white choice, and respondents would be able to express their judgment about these scientific statements as true or false (or declining to express their opinion). When a well-known scientific statement is marked as false, this means an explicit refusal to admit it as a valid premise to analyze a certain situation.

One could argue that the level of admission or rejection can also be variable, but we cannot overlook the fact that respondents are aware they are invited to make a judgment different from simply recognizing the existence of an idea. In other words, respondents know what is expected from them. The scientific statement about the age of the planet is widely known, both from the scientific and the fundamentalist religious points of view. When ticking "false", the respondents are confirming their rejection of a statement known to be accepted by a wide scientific community. A weak disagreement in a Likert scale should not be taken as equivalent as rejection, as one could say that there is some disagreement regarding the presented cipher, as a professional paleontologist probably would argue about the precise age of our planet. This disagreement cannot be taken as similar to the one from a Young Earth creationist, but in a Likert scale, both would have similar scores (1 and 2 out of 5 points in the Likert scale).

Thus, in such cases, weak and strong disagreement may represent completely different attitudes, but with similar numerical values, undermining the measure's precision. The option to

change the instrument's true/false options could enhance measurability, a critical aspect of survey sampling [58]. Thus, contrary to some of our previous pieces of research, we did not adopt Likert scale items in the questionnaire used in this study, which is presented below.

## Materials and methods

### Survey instrument

We sought to measure evolution acceptance based on five clear statements related to the evolutionary theory: Earth age, the fossil record, common ancestry, and the origin of human beings. We included an item about understanding, related to deep time, as it has been considered a relevant barrier to evolution acceptance [59], while statements about human evolution are highly controversial even amongst biology teachers [60]. As mentioned before, the section of evolution provided scientific statements as a separated set of items as part of a longer questionnaire within the Relevance of Science Education (ROSE) project [see 56]. One seminar was held in Venice (Italy) and another in Brasília (Brazil) in the year 2012, bringing together researchers from both countries for the construction of the research instrument. Collaborative arrangements were defined for conducting joint data collection using the same instrument, whose final wording was defined after fieldwork and validation with students from two high schools in the following year and applied in early 2014 in both countries [54].

Eight items related to evolution were offered to the students under a command line asking to simply tick each of the following statements as true or false, and a gentle refusal to answer the question ("I wouldn't know how to put it"). The Italian expression ("non saprei dire") was considered similar to the Portuguese translation adopted in Brazil ("não saberia dizer"). The statements are presented below, according to their identification in Section G of the questionnaire, which was then called the "SAPIENS Barometer" of the second round of the ROSE project in Italy and Brazil. The items read as follows:

- G75—The formation of our planet occurred some 4.5 billion years ago;

- G76—Fossils are evidence of living beings that lived in the past;

- G77—Present-day species of animals and plants originated from other species of the past;

- G78—Evolution occurs in both plants and animals;

- G79—Humans are descended from other primate species;

- G80—The human species has inhabited planet Earth in the last 100,000 years;

- G81—Different organisms may have a common ancestor;

- G83—The first humans were prey to carnivorous dinosaurs.

As item G82 was inadvertently absent in the Italian questionnaire, it was not considered here for comparisons between Brazilians and Italians.

### Sample design and research ethics

Sample design is basically composed of two aspects: a selection process, defining the rules and operations leading to the definition of clear targets in a wide population, and the estimation process, leading to sample estimates of population values [58]. These two aspects were standardized in both countries since the two research teams were taking part in the same international project above mentioned [56]. This international project targeted students who were beginning high school level for several reasons; one of them is related to the fact that at this age

level, moral development is already completed in most cultures, predicting long-term behavior [61]. The study was conducted using stratified sampling both in Italy and Brazil.

The Brazilian sample was designed as a sub-sample of the 2009 OECD-PISA sample and considered characteristics such as the offering of high school levels, the administrative organization of the school (public or private), the location (rural or urban, including all capitals and cities in the interior of each state), and the Human Development Index (HDI) of the state. There were three explicit stratification variables (state, grade status, and certainty), with a total of 82 explicit strata, and three implicit stratification variables (origin of funding, urban/rural, and HDI level) [62]. The PISA sample comprised 976 schools from the 27 units of the Brazilian Federation, from which 120 schools were randomly drawn, with implicit stratification to allow all federation units to be part of the subsample. This total included 20 schools for replacement.

Parcels with paper questionnaires and instructions were sent by ordinary mail to 100 schools located in 87 different municipalities all around the country after phone/email contact with the schools' principals [63]. Differences from the PISA procedures were explained. They were to select one classroom from the first year of high school ("*primeiro ano do Ensino Médio*") by a random drawing. A reasonable number of schools (n = 78) returned the paper questionnaires after a few weeks, with short reports describing how the process was carried out and a signed letter (see below). After reading the report, the questionnaires were scanned for machine-reading, leading to a digital database with answers for 96 items and personal and family details, such as age, gender, religious membership, and socioeconomic indicators. The number of students reached 2,404 from 72 municipalities of 26 federative units of the country (see Appendix 1 in Oliveira [63, p. 288–9]).

The sampling was carried out by the EDEVO-Darwin Research Nucleus in the first half of 2014, following all the applicable ethical procedures. The contact with school principals included an explanation of how the school was selected, differences from OECD/PISA technical procedures, the voluntary nature of participation, students' right to withdraw at any time with no penalties, and the maintenance of the confidentiality of school participation and guarantee of student anonymity. School principals demonstrated their accordance with the ethical procedures, sending back the parcel with the questionnaires to the postal address of the research nucleus by ordinary mail, together with a signed letter of informed consent (see Appendix A in Pinafo [64, p. 437–442]). The expected risks and benefits were explained, and mutual advantages were effectively achieved [65]. The database built by the EDEVO-Darwin Research Nucleus was already anonymized, and, in addition, the schools were recoded to prevent identification. Very similar procedures were adopted by Observa, the institution that constructed the Italian database in the same time-frame (see below).

The Italian sample was designed following the stratification model, with random drawings from the universe of secondary schools (n = 2,862), with two explicit stratification variables (geographical area, Grade 2 middle school status), with a total of 18 explicit strata, and two implicit stratification variables (school type, town type). The official list included high schools and enrolled students in Grade 2 of middle school ("secondo anno della Scuola Media di Secondo Grado"), in the year 2013/2014 [66], which corresponds to the same age level of the Brazilian sample.

The sampling in Italy was proportional to the number of students in the grade that year in each one of the geographical areas in which the country is divided, as follows: North-West (25%), North-East (18%), Center (18%), South (27%), and Islands (12%). A random selection of 100 schools followed the above-mentioned proportions. Instead of having the urban/rural school-type stratification, standard sampling procedures adopted in similar surveys in Italy recommend dividing the sample into two parts, drawing half of the schools from the group of regional major towns ("comuni capoluogo"), and the other half from all other municipalities

("territorio provinciale") [67]. The questionnaires were sent to 103 schools after 14 replacements were randomly selected, having received materials in return from 99 schools from 88 municipalities from all geographical areas of the country, comprising 3503 students. Differently from what happened in Brazil, the procedures for database entry from the paper questionnaires were done manually, with no machine-reading.

In Italy, the questionnaires were delivered to two classrooms for each selected school. The teachers explained to the students the objectives of the ROSE survey in each classroom by specifying how to fill it out and ensuring that the questionnaire was completely anonymous, along with all the other conditions in compliance with Italian legislation at the time (2014) [68]. Subsequently, after a brief introduction to the questionnaire, the students filled it out in paper or in the school computer lab, under teacher supervision, assuring they were working individually. Regulations for research on education in Italy are similar to the Brazilian ones and were followed, with no identification of the students or schools in the public databank, which was merged with the Brazilian one.

The aggregated databank has been used for independent analyses, generating doctoral theses at different institutions and in several publications [46,63,64,69]. Using an anonymous database with public access, the authors did not submit the statistical analysis protocol to an Institutional Research Review Board in advance.

## Data collection

The survey took place between March and May 2014 in both countries. Paper-and-pencil questionnaires were used in both countries, as well as electronic forms in some schools in Italy, all under absolute anonymity. The instrument consisted of a section regarding personal data (gender, age, living place), some socioeconomic questions, 73 Likert scale items, and the true/false section on evolution (G Section), which is the focus of the present study (see Annex A in Oliveira [63]).

The students had roughly the same age level in both countries, being around 15 years old. Valid cases comprised only students from 14 to 16 years old. There was a slight difference in the gender proportion in the two countries, as girls slightly outnumbered boys in Brazil (55%) and Italy (52%). The declaration of religion is a sensitive issue, and each country had a different presentation of the question. In Brazil, there was a direct question about religious affiliation (yes/no) and 18 options for answers (Catholic/Orthodox/Evangelical denominations (10 options), Jewish/ Buddhist/African denominations (2 options), Kardecist, and an "other" option). Some 12% of Brazilian students declared not to follow any religion. A high proportion of Brazilian religious students declared to be Catholic (56%); the largest non-Catholic Christian group was the Pentecostal Christians (21%), followed by Mission Evangelicals (10%), which were composed mainly of Baptists (6%) and Seventh-Day Adventists (2%). The number of Orthodox, Lutheran, and Anglican Christians was low, less than 1%. Other religions amounted to some 13% [63].

In the Italian context, this section asked a general question about religion or transcendent philosophy. Two-thirds of the students declared to follow the Catholic religion (67%), and 22% declared not to follow any religion or transcendent philosophy. Of the respondents, 3% were non-Catholic Christians, and roughly the same were "other religions" (3.5%) [48]. Therefore, the total number of Catholics in our sample from the two countries was very high, reaching over 3,000 valid cases. The number of Brazilian non-Catholic Christians was reasonable, reaching 549 valid cases. However, they composed a very conservative group compared to, for instance, the group in Britain who responded to a survey about evolution [24], where the proportion of Anglicans and Lutherans was far higher than Pentecostal Christians and Independent Evangelical Christians, the two main groups in Brazil.

## Statistical and numerical analyses

All analyses were performed using the statistical software IBM SPSS Statistics 26. We used Chi-square statistics to determine if there were statistically significant differences between the answers from the two countries, different religious affiliations in the same country, and between Catholics of the two countries. Column proportion tests were also performed, testing whether, for each row, the proportion of respondents in one column was significantly different from the proportion in the other column. For each item, a global significance level of 5% was adopted. As we were working with multiple comparisons, the significance level for each individual test was adjusted according to the Bonferroni correction method [70] to maintain a global significance of 5%. Additional tables with general results can be found in the Supporting Information (S1 and S2 Tables). As the results show very significant differences, we tackled the problem of association among a set of variables using a multivariate method, multiple correspondence analysis (MCA) [71]. Another set of tables was provided, excluding the neutral option, and allowing for an exploratory graph (biplot) to be drawn with the true/false answers from seven items with significant values (G75, G76, G77, G79, G80, G81, and G83) from the samples of the two countries. The reasons to exclude G78, taking its results as outliers, are presented in the Supporting Information (S1 File). For this analysis, the normalization method followed the main variable, using default procedures of IBM SPSS Statistics v.26. Each item was recorded according to the individuals' nationality, religion, and option (true/false), generating eight labels:

- 1 = BC + (Brazilian Catholic who answered "True");

- 2 = BC—(Brazilian Catholic who answered "False");

- 3 = BNC + (Brazilian non-Catholic Christian who answered "True");

- 4 = BNC—(Brazilian Christian non-Catholic who answered "False");

- 5 = IC + (Italian Catholic who answered "True");

- 6 = IC- (Italian Catholic who answered "False");

- 7 = INC + (non-Catholic Christian Italian who answered "True");

- 8 = INC- (Italian Christian non-Catholic who answered "False").

This led to a generalization showing four clear groups of students separated by both religion and nationality. The total number of students in the two countries who declared their religion was 3,944, but 63 had missing values in all seven items and were not considered; therefore, a total of 3,881 cases were processed. S3 Table in the Supporting Information shows the case-processing summary, S4 Table presents the numbers of every religious group in the two countries, and S5 Table shows the degree of association of each item with the two dimensions of the analysis. Supporting Information (S2 File) brings a detailed study of the MCA analysis (S6.1 to S6.10 Table in S2 File), leading to the conclusion that there is no evidence that the MCA results have any significant bias when comparing the responses of valid active cases with those with missing values. In the same file, a link provides access to the databank and the SPSS syntaxes.

We then estimated the differences between the Christian denominations (Catholics and non-Catholics) within the same country (Brazil), such that BCat—BNCat = ΔBChr. In addition, we estimated differences within the same religion (Catholics) between countries (Brazil and Italy), such that ICat—BCat = ΔCat. Thereupon, we created an intercultural index (IntcI)

to compare these differences, such that

$$IntcI = |\Delta\ Cat|/|\Delta\ BChr| \tag{1}$$

If the intercultural index was higher than 1 (IntcI>1), this meant that the Brazilian and Italian Catholics had more differences amongst themselves than among Brazilian Catholics and Brazilian non-Catholics Christians. This indicated that sociocultural factors played a more important role in the acceptance of evolution than religion itself. If the index was lower than 1 (IntcI<1) this indicated the opposite, that religion played a major role.

## Results

Previous analyses of this sample already showed significant differences between Brazilian and Italian students [54,63]. It was clear that when human evolution was specifically the focus, the level of acceptance of evolution was lower in both countries. The same broad results were found in places where the context is particularly important for the historical development of the theory of evolution, such as the Galapagos Islands [72,73]. These analyses suggest that religion plays a considerable role in the level of support for evolution. S1 and S2 Tables in the Supporting Information provide a wider view, showing that the Brazilian and Italian students have very significant differences (p< 0.0005) regarding their answers to all items.

Our main objective was to verify whether the answers in the same religious group (Catholics) were more similar in the two countries than among different denominations (Catholic and non-Catholic Christians) within the same country (Brazil). Thus, we compared students from the two countries who share the same religion. Table 1 presents the results for each item comparing Brazilian and Italian Catholics.

Proportion column tests were performed with the results of Table 1, and very significant differences were found in almost all lines (Table 2), confirming the distinct pattern shown in Table 1.

We then investigated the differences within Brazil, comparing the two groups of Christians (Table 3), with further analysis of the proportion column test (Table 4).

In the first analysis of the Brazilian sample (Table 3), we found four items (G78, G80, G81, and G83) with no significant differences between Brazilian Catholics and Brazilian non-Catholic Christians. Two of them, G80 ("Human species has inhabited planet Earth in the last 100,000 years"), and G81 ("Different organisms may have a common ancestor") had roughly the same answers in the two groups, with a high proportion of the "I would not know how to put it". Answers to the last item, G83 ("The first humans were prey to carnivorous dinosaurs") are evidence of a considerable lack of understanding about geological time in both groups in Brazil.

The only truly unexpected result with Intercultural Index (IntcI) values was found with item G80, which states "Human species has inhabited planet Earth in the last 100,000 years", as the level of agreement was lower among Italian Catholics than Brazilian ones (Tables 1 and 2). The level of disagreement (around 30%) to this statement was higher than agreement (around 20%), and around 50% of Catholic Italians preferred not to answer it. In Brazil, the situation was also unexpected, as there were no statistically significant differences between different religious denominations. The abstention of Brazilian religious students was also high (around 56%) and the level of disagreement in both groups was low (15%) compared to the level of agreement (almost 30%). Therefore, there were significant differences between Brazilian and Italian Catholics, but opposite to what was expected.

In this case, Italian youngsters could have taken the statement as an exact cipher to be confirmed or not, due to a refined lexicon interpretation derived from literal translation. Well-

**Table 1. Results by country (students declared as Catholics).**

| | | | Brazil (Catholics) | Italy (Catholics) | p-value (Chi-Square) |
|---|---|---|---|---|---|
| **G75** (Planet age) | **True** | **N** | 465 | 1659 | **< 0.0005** |
| | | **Perc. (%)** | 48.34% | 73.57% | |
| | **False** | **N** | 77 | 232 | |
| | | **Perc. (%)** | 8.00% | 10.29% | |
| | **Don't Know** | **N** | 420 | 364 | |
| | | **Perc. (%)** | 43.66% | 16.14% | |
| | **Total** | **N** | 962 | 2255 | |
| | | **Perc. (%)** | 100.00% | 100.00% | |
| **G76** (Fossils) | **True** | **N** | 821 | 2175 | **< 0.0005** |
| | | **Perc. (%)** | 84.81% | 95.10% | |
| | **False** | **N** | 50 | 49 | |
| | | **Perc. (%)** | 5.17% | 2.14% | |
| | **Don't Know** | **N** | 97 | 63 | |
| | | **Perc. (%)** | 10.02% | 2.75% | |
| | **Total** | **N** | 968 | 2287 | |
| | | **Perc. (%)** | 100.00% | 100.00% | |
| **G77** (Emergence of species) | **True** | **N** | 641 | 1946 | **< 0.0005** |
| | | **Perc. (%)** | 66.63% | 85.24% | |
| | **False** | **N** | 95 | 118 | |
| | | **Perc. (%)** | 9.88% | 5.17% | |
| | **Don't Know** | **N** | 226 | 219 | |
| | | **Perc. (%)** | 23.49% | 9.59% | |
| | **Total** | **N** | 962 | 2283 | |
| | | **Perc. (%)** | 100.00% | 100.00% | |
| **G78** (Evolution in plants and animals) | **True** | **N** | 656 | 1597 | 0.101 |
| | | **Perc. (%)** | 68.19% | 70.45% | |
| | **False** | **N** | 95 | 173 | |
| | | **Perc. (%)** | 9.88% | 7.63% | |
| | **Don't Know** | **N** | 211 | 497 | |
| | | **Perc. (%)** | 21.93% | 21.92% | |
| | **Total** | **N** | 962 | 2267 | |
| | | **Perc. (%)** | 100.00% | 100.00% | |
| **G79** (Primate origin of human beings) | **True** | **N** | 466 | 1926 | **< 0.0005** |
| | | **Perc. (%)** | 48.54% | 84.73% | |
| | **False** | **N** | 206 | 133 | |
| | | **Perc. (%)** | 21.46% | 5.85% | |
| | **Don't Know** | **N** | 288 | 214 | |
| | | **Perc. (%)** | 30.00% | 9.41% | |
| | **Total** | **N** | 960 | 2273 | |
| | | **Perc. (%)** | 100.00% | 100.00% | |
| **G80** (Age of the human species) | **True** | **N** | 265 | 451 | **< 0.0005** |
| | | **Perc. (%)** | 27.72% | 19.97% | |
| | **False** | **N** | 149 | 668 | |
| | | **Perc. (%)** | 15.59% | 29.58% | |
| | **Don't Know** | **N** | 542 | 1139 | |
| | | **Perc. (%)** | 56.69% | 50.44% | |
| | **Total** | **N** | 956 | 2258 | |
| | | **Perc. (%)** | 100.00% | 100.00% | |

*(Continued)*

**Table 1.** (Continued)

| | | | Brazil (Catholics) | Italy (Catholics) | p-value (Chi-Square) |
|---|---|---|---|---|---|
| **G81** (Common ancestor) | **True** | N | 296 | 1370 | **< 0.0005** |
| | | Perc. (%) | 31.09% | 60.25% | |
| | **False** | N | 168 | 260 | |
| | | Perc. (%) | 17.65% | 11.43% | |
| | **Don't Know** | N | 488 | 644 | |
| | | Perc. (%) | 51.26% | 28.32% | |
| | **Total** | N | 952 | 2274 | |
| | | Perc. (%) | 100.00% | 100.00% | |
| **G83** (Human/dinosaur coexistence) | **True** | N | 220 | 293 | **< 0.0005** |
| | | Perc. (%) | 22.89% | 12.87% | |
| | **False** | N | 346 | 1378 | |
| | | Perc. (%) | 36.00% | 60.52% | |
| | **Don't Know** | N | 395 | 606 | |
| | | Perc. (%) | 41.10% | 26.61% | |
| | **Total** | N | 961 | 2277 | |
| | | Perc. (%) | 100.00% | 100.00% | |

General profile of answers in Brazil and Italy, with Chi-square tests (religious group: Catholic).

**Table 2. Proportion column tests by country (religious group: Catholic).**

| | | Brazil | Italy |
|---|---|---|---|
| | | (A) | (B) |
| **G75** (Planet age) | True | | A (< 0.0005) |
| | False | | A (0.044) |
| | Don't Know | B (< 0.0005) | |
| **G76** (Fossils) | True | | A (< 0.0005) |
| | False | B (< 0.0005) | |
| | Don't Know | B (< 0.0005) | |
| **G77** (Emergence of species) | True | | A (< 0.0005) |
| | False | B (< 0.0005) | |
| | Don't Know | B (< 0.0005) | |
| **G78** (Evolution in plants and animals) | True | | |
| | False | B (0.035) | |
| | Don't Know | | |
| **G79** (Primate origin of human beings) | True | | A (< 0.0005) |
| | False | B (< 0.0005) | |
| | Don't Know | B (< 0.0005) | |
| **G80** (Age of the human species) | True | B (< 0.0005) | |
| | False | | A (< 0.0005) |
| | Don't Know | B (0.001) | |
| **G81** (Common ancestor) | True | | A (< 0.0005) |
| | False | B (< 0.0005) | |
| | Don't Know | B (< 0.0005) | |
| **G83** (Human/dinosaur coexistence) | True | B (< 0.0005) | |
| | False | | A (< 0.0005) |
| | Don't Know | B (< 0.0005) | |

**Table 3. Answers of Brazilian Christians, with Chi-square tests.**

| | | | Catholic | Non-Catholic Christian | Total | p-value (Chi-Square) |
|---|---|---|---|---|---|---|
| G75 (Planet age) | True | N | 465 | 225 | 690 | **0.002** |
| | | Perc. (%) | 48.34% | 40.98% | 45.67% | |
| | False | N | 77 | 69 | 146 | |
| | | Perc. (%) | 8.00% | 12.57% | 9.66% | |
| | Don't Know | N | 420 | 255 | 675 | |
| | | Perc. (%) | 43.66% | 46.45% | 44.67% | |
| | Total | N | 962 | 549 | 1511 | |
| | | Perc. (%) | 100.00% | 100.00% | 100.00% | |
| G76 (Fossils) | True | N | 821 | 432 | 1253 | **0.012** |
| | | Perc. (%) | 84.81% | 78.83% | 82.65% | |
| | False | N | 50 | 42 | 92 | |
| | | Perc. (%) | 5.17% | 7.66% | 6.07% | |
| | Don't Know | N | 97 | 74 | 171 | |
| | | Perc. (%) | 10.02% | 13.50% | 11.28% | |
| | Total | N | 968 | 548 | 1516 | |
| | | Perc. (%) | 100.00% | 100.00% | 100.00% | |
| G77 (Emergence of species) | True | N | 641 | 324 | 965 | **< 0.001** |
| | | Perc. (%) | 66.63% | 59.23% | 63.95% | |
| | False | N | 95 | 93 | 188 | |
| | | Perc. (%) | 9.88% | 17.00% | 12.46% | |
| | Don't Know | N | 226 | 130 | 356 | |
| | | Perc. (%) | 23.49% | 23.77% | 23.59% | |
| | Total | N | 962 | 547 | 1509 | |
| | | Perc. (%) | 100.00% | 100.00% | 100.00% | |
| G78 (Evolution in plants and animals) | True | N | 656 | 363 | 1019 | 0.077 |
| | | Perc. (%) | 68.19% | 66.61% | 67.62% | |
| | False | N | 95 | 74 | 169 | |
| | | Perc. (%) | 9.88% | 13.58% | 11.21% | |
| | Don't Know | N | 211 | 108 | 319 | |
| | | Perc. (%) | 21.93% | 19.82% | 21.17% | |
| | Total | N | 962 | 545 | 1507 | |
| | | Perc. (%) | 100.00% | 100.00% | 100.00% | |
| G79 (Primate origin of human beings) | True | N | 466 | 161 | 627 | **< 0.0005** |
| | | Perc. (%) | 48.54% | 29.38% | 41.58% | |
| | False | N | 206 | 225 | 431 | |
| | | Perc. (%) | 21.46% | 41.06% | 28.58% | |
| | Don't Know | N | 288 | 162 | 450 | |
| | | Perc. (%) | 30.00% | 29.56% | 29.84% | |
| | Total | N | 960 | 548 | 1508 | |
| | | Perc. (%) | 100.00% | 100.00% | 100.00% | |
| G80 (Age of the human species) | True | N | 265 | 160 | 425 | 0.778 |
| | | Perc. (%) | 27.72% | 29.41% | 28.33% | |
| | False | N | 149 | 84 | 233 | |
| | | Perc. (%) | 15.59% | 15.44% | 15.53% | |
| | Don't Know | N | 542 | 300 | 842 | |
| | | Perc. (%) | 56.69% | 55.15% | 56.13% | |
| | Total | N | 956 | 544 | 1500 | |
| | | Perc. (%) | 100.00% | 100.00% | 100.00% | |

(*Continued*)

**Table 3.** (Continued)

| | | | Catholic | Non-Catholic Christian | Total | p-value (Chi-Square) |
|---|---|---|---|---|---|---|
| **G81** (Common ancestor) | **True** | N | 296 | 151 | 447 | 0.327 |
| | | Perc. (%) | 31.09% | 27.86% | 29.92% | |
| | **False** | N | 168 | 108 | 276 | |
| | | Perc. (%) | 17.65% | 19.93% | 18.47% | |
| | **Don't Know** | N | 488 | 283 | 771 | |
| | | Perc. (%) | 51.26% | 52.21% | 51.61% | |
| | **Total** | N | 952 | 542 | 1494 | |
| | | Perc. (%) | 100.00% | 100.00% | 100.00% | |
| **G83** (Human/dinosaur coexistence) | **True** | N | 220 | 105 | 325 | 0.105 |
| | | Perc. (%) | 22.89% | 19.44% | 21.65% | |
| | **False** | N | 346 | 222 | 568 | |
| | | Perc. (%) | 36.00% | 41.11% | 37.84% | |
| | **Don't Know** | N | 395 | 213 | 608 | |
| | | Perc. (%) | 41.10% | 39.44% | 40.51% | |
| | **Total** | N | 961 | 540 | 1501 | |
| | | Perc. (%) | 100.00% | 100.00% | 100.00% | |

**Table 4. Proportion column tests (Brazilian Catholics and non-Catholics).**

| | | Catholic | non-Catholic Christian |
|---|---|---|---|
| | | (A) | (B) |
| **G75** (Planet age) | True | B (0.006) | |
| | False | | A (0.004) |
| | Don't Know | | |
| **G76** (Fossils) | True | B (0.003) | |
| | False | | |
| | Don't Know | | A (0.039) |
| **G77** (Emergence of species) | True | B (0.004) | |
| | False | | A ($< 0.0005$) |
| | Don't Know | | |
| **G78** (Evolution in plants and animals) | True | | |
| | False | | A (0.029) |
| | Don't Know | | |
| **G79** (Primate origin of human beings) | True | B ($< 0.0005$) | |
| | False | | A ($< 0.0005$) |
| | Don't Know | | |
| **G80** (Age of the human species) | True | | |
| | False | | |
| | Don't Know | | |
| **G81** (Common ancestor) | True | | |
| | False | | |
| | Don't Know | | |
| **G83** (Human/dinosaur coexistence) | True | | |
| | False | | |
| | Don't Know | | |

informed students about human paleontological details may know that the human species appeared longer than 100,000 years before present, and may have decided to take the statement as false, or rather to refrain from giving a clear opinion. On the Brazilian side, the level of understanding about paleontology was lower, as shown by the answers of item G83, leading to a decreased level of criticism upon the statement, but with higher abstention. The anomalous behavior of the item can also be noticed in the two dimensions of the biplot graph, as the degree of association of the item with the dimensions of the analysis had the lowest value (see S5 Table in the Supporting Information).

Brazilian Catholics showed a significantly lower level of knowledge about geological time than the Italians (Tables 1 and 2). However, there were no significant differences between the two Christian denominations in Brazil (Tables 3 and 4). Thus, the reason for Italians to disagree with the statement of G80 could be different from that of the Brazilians, showing a polarized pattern due to differences in understanding that are related to education. Therefore, the answers could have resulted from their different level of information, following the general pattern of the two samples, and the item was not discarded.

The results tend to show that those who follow the very same religion, listening to the teaching of the very same Catholic religious authorities in different countries, had bigger differences regarding evolution acceptance from people who were brought up in the same socio-cultural environment but followed different religious teachings. In addition, differences in understanding tended to follow the same pattern. This hypothesis found additional support with the multiple correspondence analysis (MCA), analyzing the "true" and "false" options in the two religious denominations in the two countries in all items. The overall picture can be seen in Fig 1.

The graph of joint categories (biplot) displays the average scores of the two dimensions for individuals belonging to the four categories. The origin of the graph represents an individual who belongs to the most frequent category of each variable; therefore, profiles that are far from the origin indicate groups that are outside of the general pattern of the data. There are four clear groups of respondents, showing that Catholics do not form a homogenous group. On the contrary, Brazilian and Italian Catholics are far from being in the same general pattern of answers, confirming what is seen in Tables 1 and 2.

The relative numbers of "true" answers were then compared and the module of the differences between Catholics in the two countries originated the delta Catholics ($\Delta$**Cat**), as well as the differences between Christian denominations in the same country ($\Delta$**BChr**) used in Eq 1. The results can be seen in Table 5.

These results were plotted as shown in Fig 2, which gives a better idea of the general results in terms of the relative importance of religion (orange polygon) and wider socio-cultural factors within each country (blue polygon). The religion borderline assumes that there should be small differences between the same Christian denomination (Christian Catholic) in the two countries, and higher differences between the two different Christian denominations. Therefore, the results appear plotted inside the orange polygon. Differences in the understanding of geological time followed the same pattern, with a very high score (G83 IntcI = 7.41), but they were not included in the same figure as they are not related to acceptance (see text).

## Discussion

Research on evolution education faced a big challenge after the surprising findings showing that different instruments lead to different rates of agreement/rejection of evolution in the same student population [6]. We tried to show that some instruments could have had problems in terms of their methodology, implying distortions for the numerical analysis, as

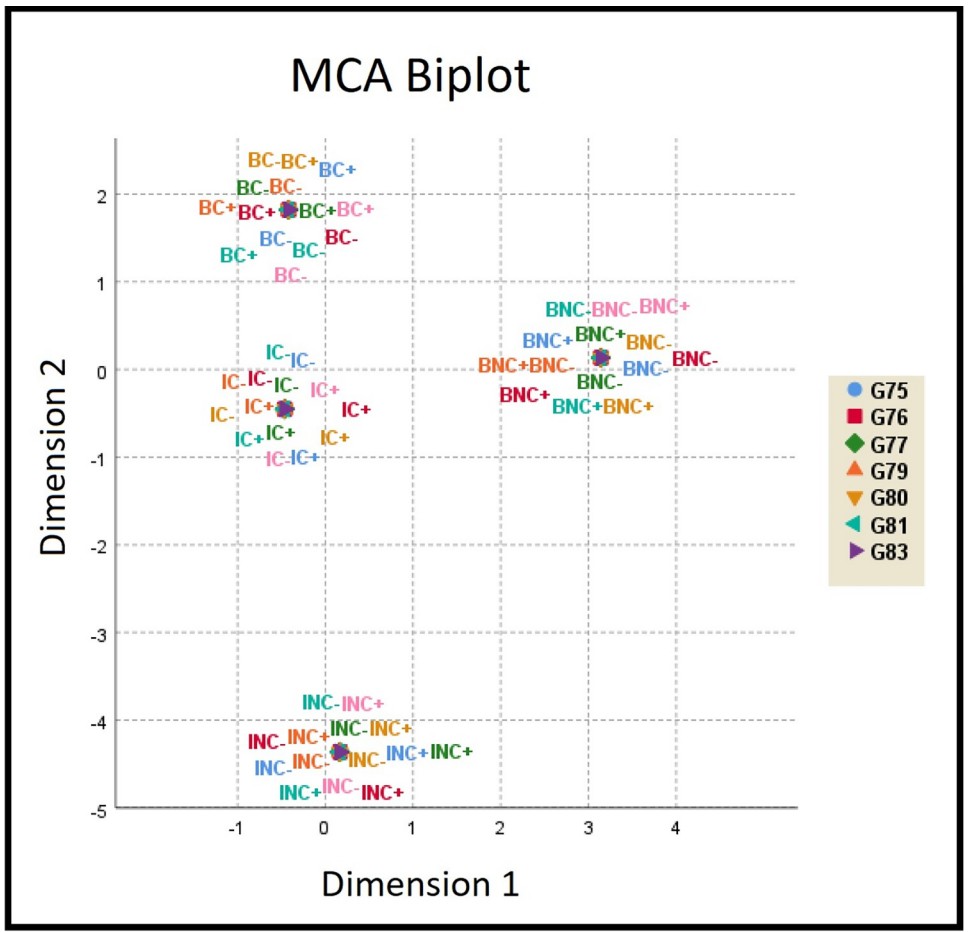

**Fig 1. Multiple correspondence analysis biplot considering evolution acceptance and understanding.**

**Table 5. Intercultural Index (IntcI) related to evolution acceptance.**

| | BCat | BNCC | ICat | |ΔCat| | |ΔBChr| | IntcI |
|---|---|---|---|---|---|---|
| **G75** (Planet age) | 0.483 | 0.409 | 0.736 | 0.252 | 0.074 | **3.43** |
| **G76** (Fossils) | 0.848 | 0.788 | 0.911 | 0.103 | 0.059 | **1.72** |
| **G77** (Emergence of species) | 0.666 | 0.592 | 0.852 | 0.186 | 0.074 | **2.51** |
| **G79** (Primate origin of human beings) | 0.485 | 0.294 | 0.847 | 0.362 | 0.192 | **1.89** |
| **G80** (Age of the human species) | 0.277 | 0.294 | 0.199 | 0.077 | 0.017 | **4.58** |
| **G81** (Common ancestor) | 0.311 | 0.279 | 0.602 | 0.292 | 0.032 | **9.02** |

Table with values of relative proportion between |ΔCat| (differences of agreement between Italian Catholics (ICat) and Brazilian Catholics (BCat)) and |ΔBChr| (differences of agreement between Brazilian Catholics (BCat) and Brazilian non-Catholic Christians (BNCC) about well-known scientific statements.

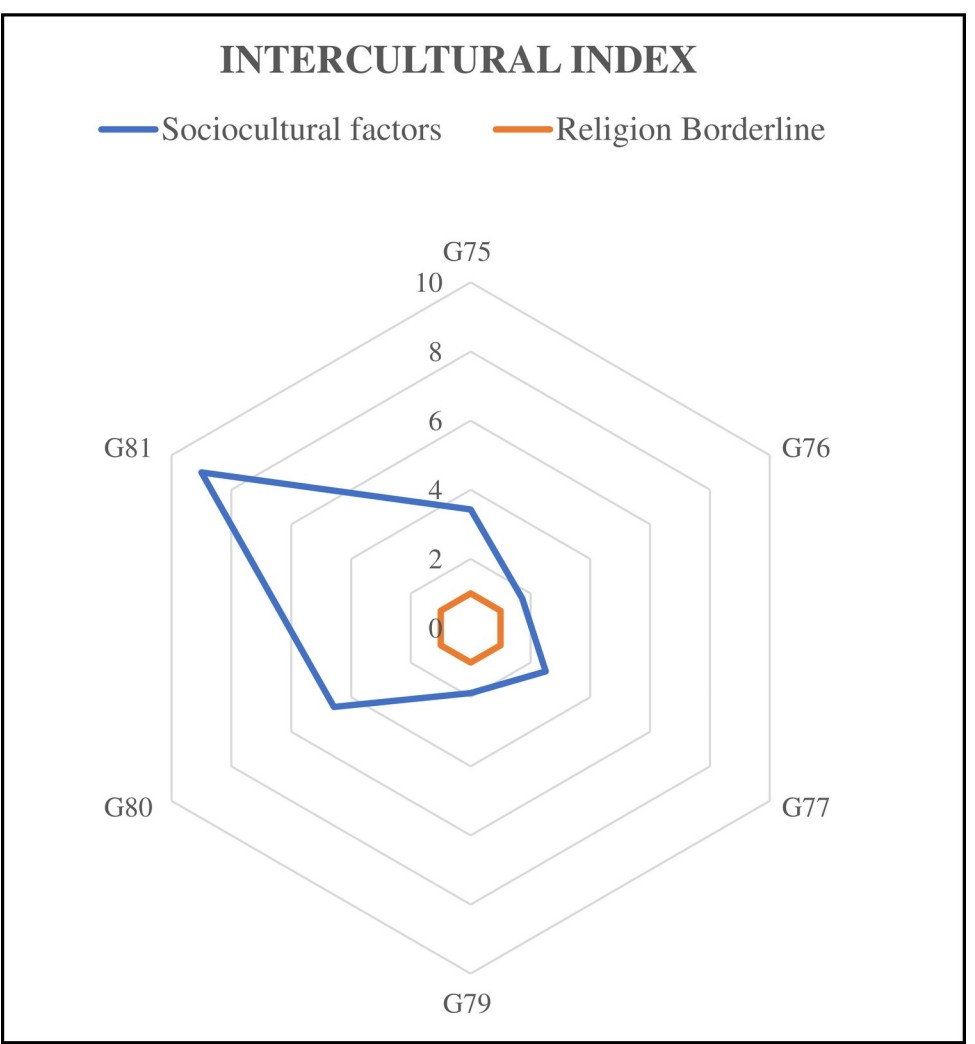

**Fig 2. Data from Table 5 graphically displayed.** The area of the small polygon corresponds to the expected result in the case that all Christians had similar answers in both countries. The blue polygon indicates greater differences between Catholics in both countries than differences between the two groups of Christians in the same country (Brazil).

measurements could result in some bias. These distortions can explain, at least in part, the mentioned puzzle. In addition, there is not consensus in the educational community about how important religion is for the acceptance of evolution.

We presented a new instrument, stating some principles on how to evaluate students' opinions concerning evolution, asking clear positions about well-known statements on the subject and using nationwide samples of students of the same age (around 15 years old) and absolute anonymity.

The results show a clear picture of the two countries, which is in line with recent data [45], but with a specific focus on two groups of Christians, with significant differences between Catholics in Italy and Brazil. Moreover, the differences were larger between students from the same Christian denomination (Roman Catholic) but different countries than from different Christian denominations in the same country (Brazil). The MCA found clear distinct groups of respondents. However, this is simply an exploratory data technique, and the final picture should be seen in a purely descriptive way, without the power of indicating a precise output of

a statistical test. As the number of valid cases involved was not very high (n = 3,881), relative distances that can be inferred should be viewed with caution (see S5 Table in Supporting Information).

Item G79 ("Humans are descended from other primate species") was based on the well-known instrument "MATE", keeping the same wording in the revised version [8] for the statement about humans, revealing difficulties of evolution in 34 countries [2]. Many studies have shown that when the human species is concerned, opinions about evolution tend to be extreme, with some people considering evolution for all organisms except our species [60]. This is perhaps the best item to test the hypothesis we have been examining, as the number of respondents of both countries was not low (n = 3,117). There was a clear shift between the Roman Catholics, with a high acceptance (84.7%) and low rejection (5.9%) among Italians, whereas in Brazil, the situation was very different, with a significantly lower acceptance (48.5%) and higher rejection (21.5%) (Table 1). Within Brazil, differences between Christian denominations were lower (Table 3).

Tables 3 and 4 show that G81 ("Different organisms may have a common ancestor") had no significant statistical differences between religious denominations in the same country, and Tables 1 and 2 show very significant differences between Roman Catholics in the two different countries. As these differences refer to the very same religious denomination, which has very similar values and beliefs in the two countries, it would be reasonable to expect the opposite situation, with no significant differences among people who follow the same religious teachings from an early age.

The level of understanding about geological time was very different between Italian and Brazilian Catholics, but a similar level of low knowledge was found in both Christian denominations in Brazil. This not only adds evidence to the importance of understanding this important concept for evolution acceptance, but also shows that education and religion have a strong link with nationality, adding evidence to previously published studies [6,24]. In our case, Italian Catholics had a higher acceptance as well as a higher level of understanding of evolution than Brazilian ones. Independent recent research obtained similar results, showing that Catholic and Evangelical Brazilian university students had low levels of knowledge about evolution [74].

Notwithstanding this general picture found after the MCA, additional evidence in numerical terms supports the idea that religion alone does not play a major role in the acceptance of evolution, with the relative size of differences obtained by the Intercultural Index (IntcI). It tended to show small numbers (IntcI<1) in cases where religion played a major role, but the results point in the exact opposite direction, as acceptance was far more similar between different religions in the same country than between Catholics in the two different countries.

All IntcI values were higher than 1, reaching values as high as 9, as in the case of item G81, which states that different organisms may have a common ancestor. This can be regarded as a clear-cut, widely known, scientific principle of common ancestry, and a basic principle for biological evolution. The calculated Intercultural Index for item G79 ("Humans are descended from other primate species") was high (IntcI = 1.889), showing that differences within Catholics were almost twice as high as those between them and non-Catholic Christians in the same country (Brazil).

The general picture shows that religion plays a less important role in the acceptance of evolution than nationality, which should be regarded as the wide socio-cultural environment, including religion and education. Children develop a range of different worldviews, which can affect their position towards science and scientific statements [75]. However, the acceptance of evolution cannot be seen as a simple output of a given worldview but as a complex result of several influences.

No one doubts that literalist interpretations of sacred texts can influence the learning processes of young students since early childhood. The development of understanding of central concepts can be seriously hampered by the influence of fundamentalist religious faith. The conceptual discussion about belief, faith, and acceptance [49,50,76] may be more worthwhile than creating more research instruments.

The instrument presented here could be applied in different settings, and research could be carried out with samples easier to obtain, not necessarily nationwide ones. Databanks already providing items with clear-cut scientific statements could be recoded, transforming Likert scale numbers into true/false answers. Then, the Intercultural Index presented here could objectively show the relative importance of religion for evolution acceptance. If our results are to be confirmed in the future, this could provide considerable consequences for making concrete recommendations for evolution education to religious students in conservative and anti-science environments.

It is important to keep in mind that the picture presented here is only one frame of a long film, as values are changing very rapidly, even in religious settings. For instance, the level of acceptance of evolution has improved considerably within members of some conservative Christian denominations whose teachings are informed by cultural barriers to evolution [35]. Unfortunately, change can also be noticed in the opposite direction, with conservative anti-evolution groups and religious denominations showing a growing influence around the world, Brazil included [39–41,74].

## Supporting information

**S1 Table. General profile of items by questionnaire, with Chi-square tests comparing Brazilian and Italian students (all religions comprised).**
(DOCX)

**S2 Table. Proportion column tests for each item.**
(DOCX)

**S3 Table. Summary of the multiple correspondence analysis.** MCA case processing summary.
(DOCX)

**S4 Table. Samples of students who declared their religion in Italy and Brazil.**
(DOCX)

**S5 Table. MCA matrix of discrimination measures.**
(DOCX)

**S1 File. Reasons to consider item G78 as an outlier.**
(DOCX)

**S2 File. Ancillary set of tables exploring possible bias in missing cases that were not included in the multiple correspondence analysis & link to database and SPSS syntaxes.**
(DOCX)

## Acknowledgments

Data collection was possible due to the collaboration between the Italian researchers of OBSERVA, Science in Society, and the Brazilian researchers and staff at the School of Education (FEUSP), and EDEVO-Darwin Research Nucleus (Pró-Reitoria de Pesquisa/USP). IBM SPSS v.26 statistical analyses were performed by Estéfano A. de Souza. Many people

contributed in different regards to this research, especially P. Abrantes, R. Alitto, E. Almeida, L. Bizzo, D. Borges, V. Barzotto, C. Boto, T. Forato, F. Franzolin, P. Garcia, A. Gouw, S. Martorano, P. Monteiro, M. Neira, I. Nogueira, M. Prestes, P. Prado, P. Sano, and H. Viana. We also have to thank a number of teachers, principals, students, and their families from high schools throughout Italy and Brazil for their help with data collection. Access to the PISA/Brazil school database was provided by the Brazilian Ministry of Education, INEP/DACC.

## Author Contributions

**Conceptualization:** Nelio Bizzo.

**Data curation:** Leonardo Augusto Luvison Araújo, Nelio Bizzo.

**Formal analysis:** Graciela da Silva Oliveira, Giuseppe Pellegrini, Nelio Bizzo.

**Funding acquisition:** Giuseppe Pellegrini, Leonardo Augusto Luvison Araújo, Nelio Bizzo.

**Investigation:** Graciela da Silva Oliveira, Giuseppe Pellegrini, Nelio Bizzo.

**Methodology:** Giuseppe Pellegrini, Nelio Bizzo.

**Project administration:** Giuseppe Pellegrini, Nelio Bizzo.

**Writing – original draft:** Leonardo Augusto Luvison Araújo, Nelio Bizzo.

**Writing – review & editing:** Graciela da Silva Oliveira, Giuseppe Pellegrini, Leonardo Augusto Luvison Araújo, Nelio Bizzo.

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
