## [Decision Letter · Decision Letter 0]

1 May 2022

PONE-D-22-03227Acceptance of Evolution by High School Students: is religion the key factor?PLOS ONE

Dear Dr. Bizzo,

Thank you for submitting your manuscript to PLOS ONE. After careful consideration, we feel that it has merit but does not fully meet PLOS ONE’s publication criteria as it currently stands. Therefore, we invite you to submit a revised version of the manuscript that addresses the points raised during the review process.

We look forward to receiving your revised manuscript.

Kind regards,

Norman Johnson

Academic Editor

PLOS ONE

Journal Requirements:

Additional Editor Comments (if provided):

This ms addresses an important topic - why does evolution acceptance differ? - and takes new approaches to address it.

I found the results intriguing.

Still, the reviewers noted several concerns about both the methodology of the study and the presentation in the ms. I would like to see the authors’ responses to these. think the Introduction can be compressed somewhat without loss of information. There are sections - as noted by the reviewers - that can be clarified.

In addition to the comments made by the reviewers, I think the presentation of the tables can be improved. Notably, I suggest adding a brief descriptor in addition to the G number in each table. So, G75 could be “planet age”, G76 could be “fossils”, and so on.

Reviewers' comments:

Reviewer's Responses to Questions

**Comments to the Author**

1. Is the manuscript technically sound, and do the data support the conclusions?

Reviewer #1: Yes

Reviewer #2: Partly

2. Has the statistical analysis been performed appropriately and rigorously? 

Reviewer #1: I Don't Know

Reviewer #2: No

3. Have the authors made all data underlying the findings in their manuscript fully available?

Reviewer #1: Yes

Reviewer #2: Yes

4. Is the manuscript presented in an intelligible fashion and written in standard English?

Reviewer #1: No

Reviewer #2: No

5. Review Comments to the Author

Reviewer #1: This article examines compare acceptance of evolution in high school students in Italy and Brazil, specifically looking at 1) the overall differences in acceptance between countries 2) whether level of acceptance is associated more strongly with a specific Christian denomination (Roman Catholic) in both countries, and 3) whether acceptance of evolution by both Catholic and non-Catholic Christian students within each country are more similar to each other. The authors found that the level of evolution acceptance is more similar among students within each country than among Roman Catholics in both countries. They conclude that "the sociocultural environment and level of evolutionary knowledge" is more important than the specific religious affiliation they studied (from the abstract), and that "religion playing a less important role for the acceptance of evolution that nationality" (from lines 648-649). It is an elegant study and should be published.

However, I have some suggestions that I think will strengthen the manuscript, listed below. Some are broader and more important, and some are more minor.

1. There are many areas where the article is not clearly written, and includes awkward phrasing that I am interpreting as the result of English not being the authors' first language. At times this impeded my ability to read and understand the article. As PLOS ONE does not copyedit accepted manuscripts, I would strongly recommend that the authors have a proofreader assist with this issue.

2. Throughout the paper the authors refer to "different Christian religions". I have more often heard this referred to as different Christian religious denominations, all falling under the single religion of Christianity. The authors should consider changing this accordingly.

3. There are some areas of the introduction which are repetitive (e.g., lines 98-101). I recommend that this section be condensed, and that less details of previous studies be provided (e.g., lines 112-125; the reader does not need to know the details of how the analysis was done - the shorter description of another study in lines 126-131 is a better approach).

4. Since this study focuses on acceptance of evolution among Roman Catholic students, the "explicit views on biological evolution" referenced in 146 should be described in more detail.

5. I am not sure that reference 35 in line 167 is the appropriate reference for that information.

6. The explanation of why they had the students answer their survey anonymously (lines 213-226) was interesting, but there are no citations provided for their reasoning. The addition of citations would be helpful, or being clear that this is opinion/conjecture. The same for their reasoning about why a Likert scale is not appropriate for statements expressing scientific facts, as they should not be "graded as weak or strong" (lines 243-286). Basically, I would like to see some references to survey design to help the reader have confidence that their decisions were based on best practices or previous research. I do appreciate the nuance that a Likert scale could score different approaches to an answer about the age of the planet similarly thought they might come from different places of disagreement - precision of scientific evidence vs. a non-scientific view (lines 275-280).

7. Line 358 mentions "personal and family details" - what are these? Either explain or omit this.

8. Line 610 mentions a "well-known instrument" - please include the name of this instrument.

9. Mention the question number of the survey that the results described beginning with line 626 come from.

10. I am concerned about the somewhat strident tone in lines 655-658 regarding fundamentalist religious faith - not all fundamentalist religious people deny "the objective existence of well-founded scientific facts". I also recommend removing the word "notorious" in line 672 for the same reason.

11. Which "certain conservative religions" are being referred to in lines 671-672? Be explicit.

Reviewer #2: Thank you for the opportunity to read this manuscript. This study follows up on prior work on evolution acceptance in Brazil and Italy (“Previous analysis of this sample had already shown significant differences between Brazilian and Italian students…”). The manuscript makes the claim that culture is more important than religion using Catholics in Brazil and Italy. Few would doubt that culture matters (or that religion matters) but the question is by how much and why does this matter? The study has a very large number of methodological weaknesses that need to be addressed.

Introduction

The introduction discusses how knowledge and the nature of science relate to acceptance, and yet the manuscript did not seem to measure these core variables. Differences in these variables between countries could explain the acceptance patterns discussed. This was a confusing aspect of the introduction and the study.

The introduction emphasizes “inconsistencies” and “differences” in prior research findings relating to acceptance and knowledge. However, the contexts of comparison (e.g., ages, countries), magnitudes of differences (e.g., effect sizes), meaning of the differences in light of the intended measures (e.g., instrument construct definitions), and quality of the studies (e.g., sampling, measurement statistics) being discussed were not carefully considered. A much more careful and rigorous review of the literature is needed. The current introduction does not carefully review studies with direct relevance to the populations of interest.

Line 59. The claim regarding “inconsistencies” is not supported by the literature. Studies examining different populations with different instruments and different levels of education cannot be meaningfully compared. Moreover, studies vary greatly in the quality of the measures and the rigor of the analyses. Studies of the same populations using the same or similar instruments demonstrate few if any differences, and those that were found tend to be small effect sizes (many significant differences are not meaningful, particularly when using large samples). Knowledge and acceptance have a positive association and this is widely supported. Claims of contradiction and inconsistency do not align with the preponderance of evidence in high quality studies. We can find published studies that contradict almost anything, but they tend to have significant flaws. Not all studies are equal. A RCT and a static descriptive finding are not of the same quality.

Line 66. The effect sizes for Barnes were not large, and the instruments utilized were intentionally designed to measure different aspects of evolution acceptance. It is NOT a flaw to intentionally conceptualize (and operationalize) a theoretical construct in different ways. Why would small differences using instruments that measure a construct in slightly different ways be “surprising”? Fahrenheit and Celcius are different scales of temperature–is one flawed because it produces a different measure? They are based on different operationalizations of theory.

This introduction could be fixed by reviewing findings from high-quality studies most similar in terms of religion, age, education, etc. to the new research that will be reported.

Sample

It is not clear why the two countries are being compared if the Pew studies have shown significant differences in evolution acceptance (and prior work). A stronger and clearer rationale for the comparisons (in light of the methodological issues noted above) needs to be made so that readers can see what the authors are trying to do (which is interesting).

Methods

It was not clear how religious affiliation and religiosity were measured–were they separate? These are two well-defined constructs that are different.

The methods discuss measuring evolution acceptance, but later on the discussion mentions “understanding” in several places. It was not clear if there is a knowledge measure and an acceptance measure. Or are the authors conflating acceptance and understanding? This was confusing. (e.g., “Difference of geological time understanding followed the same pattern, with a very high score”).

Line 250. It is important to consider that Likert scales by themselves are not instruments or measures. Measures are derived from the scale, and such scales pose many problems (e.g., non-ratio scaling) that can distort inferences. The way that these issues are discussed is problematic and should be revised.

Line 288. The introduction emphasizes that different instruments that measure constructs in different ways produce different results, and so it was surprising to read that a new instrument that has not been widely used was employed in this study. How, then, can the results be compared to prior work? It seems that the problems emphasized in the introduction are ignored and the authors continue the tradition of comparing patterns using different tools. How will this move research forward? The methodological approach was puzzling. Regardless, evidence in support of instrument quality needs to be expanded as it is inappropriate to interpret measures unless there is a robust body of evidence supporting them (see Mead et al. Evolution Education Outreach on instrument quality in evolution education). Please report comparable evidence types for the instrument.

The sampling methods should include information on missing data and whether it was missing at random (e.g., did students of particular groups have disproportionate missing data?). The participation rate should be mentioned in the main text.

The data have clear nesting and HLM (hierarchical linear models) would allow for the analyses of patterns within schools, regions, and countries. The current methodological approach is very unusual and so a rationale is needed for why MCA was used.

Why were the data on nationality and religion used to code responses when it would be possible to use predictor variables contained within the dataset? In other words, the analyses ignore the richness of the dataset and predispose the results to the coding scheme. This is problematic and poorly justified. Within- and between- group variation can be modeled along with interactions.

The discussion section should emphasize why these results matter. Let’s say that the results are well supported by evidence (which is by no means clear). How does this change how biologists should approach the challenge of evolution education? What are the implications for other countries? How should sociocultural factors be measured and included in models of acceptance?The discussion section should also include prior evolution education research in nonreligious contexts such as China and other international studies (please review Donnelly and Deniz Springer and other studies).

6. PLOS authors have the option to publish the peer review history of their article (what does this mean?). If published, this will include your full peer review and any attached files.

Reviewer #1: No

Reviewer #2: No

---

## [Author Response · Author response to Decision Letter 0]

1 Aug 2022

Additional Editor Comments

“Introduction can be compressed somewhat without loss of information”.

Answer: The introduction has been shortened based on reviewers' comments. Changes are specified in the comments for reviewers.

"I think the presentation of the tables can be improved. Notably, I suggest adding a brief descriptor in addition to the G number in each table. So, G75 could be “planet age”, G76 could be “fossils”, and so on”

Answer: As suggested, brief descriptors were added to the G number in each table. 

Reviewer #1

“1. There are many areas where the article is not clearly written, (…) I would strongly recommend that the authors have a proofreader assist with this issue”.

Answer: The article has undergone English language editing by a professional reviewer (MDPI). The text has been checked for correct use of grammar and common technical terms, and edited to a level suitable for reporting research in a scholarly journal. 

“2. Throughout the paper the authors refer to "different Christian religions". I have more often heard this referred to as different Christian religious denominations, all falling under the single religion of Christianity”.

Answer: In fact, this is a longstanding problematic issue. Some authors criticize the term “denomination”, for instance: “When we use the analytical term ‘religion,’ we must not think that it refers to some specific, ‘least common denominator’ super- religion or trans- religious entity, which can be partitioned into ‘denominations’. No such thing exists. All that exists in actuality are particular religions.” (Smith, 2017, p.46). However, soon after the same author recognizes that this ideal conceptual approach cannot be found in US publications, which prefer to call all Christian branches “Christian denominations”, as he himself used to call them (Smith, 2000). Roman Catholic theologians, such as John Haught, avoid the term; in his famous book “Science and Religion: From conflict to conversation” (1995), he does not use the expression “Christian denominations”. Thus, this way of referring to Christianity as a “trans-religious entity” is widely used, as by Thomas Dixon, in his book “Science and Religion: A very short introduction” (2008, Oxford University Press). Therefore, we decided to modify the expression to the most used/understandable way of defining religions in the US and changed all references of “Christian religions” into “Christian denominations”. 

 “3. There are some areas of the introduction which are repetitive (e.g., lines 98-101). I recommend that this section be condensed, and that less details of previous studies be provided”.

Answer: We condensed the discussion in the lines indicated above (lines 169-170) and elsewhere.

“4. Since this study focuses on acceptance of evolution among Roman Catholic students, the "explicit views on biological evolution" referenced in L.146 should be described in more detail”

Answer: We included a brief description of previous studies in Brazil carried out in the early 1990s, but consider that the reference that was already in the text (Oliveira & Cook, 2019) addresses appropriately the issue, which should be seen in a dynamic perspective. 

“5. I am not sure that reference 35 in line 167 is the appropriate reference for that information.” 

Answer: This reference is the Pew Research Center report, which “examines public perceptions of biotechnology, evolution and the relationship between science and religion. Data in this report come from a survey conducted across Europe, Russia, the Americas and the Asia-Pacific region from October 2019 to March 2020. Surveys were through face-to-face interviews in Russia, Poland, the Czech Republic, India and Brazil”. The title adopted as bibliographical reference is the one recommended by the report. 

“6. The explanation of why they had the students answer their survey anonymously (lines 213-226) was interesting, but there are no citations provided for their reasoning (…) Basically, I would like to see some references to survey design to help the reader have confidence that their decisions were based on best practices or previous research.”

Answer: Strong pieces of evidence published in the literature shows that anonymity reduces social desirability distortion and increases self-disclosure. This research was part of an international project originally designed back to 2004-6 relying on basic literature. Many references are available in the literature and we understand that this is a well established concept when sensitive issues are involved. However, we included two basic references regarding more recent research, addressing social desirability, including a meta-analysis comprising 51 studies that included 62 independent samples and 16,700 unique participants. 

“7. Line 358 mentions ‘personal and family details’ - what are these? Either explain or omit this”. 

Answer: We introduced one line explaining that the databank includes students’ age, gender, religious membership, and socioeconomic indicators. 

“8. Line 610 mentions a "well-known instrument" - please include the name of this instrument.”

Answer: We included the name of the “well known instrument” mentioned in the article soon after, and a recent reference, but also rephrased the first mention of it, in the introduction. 

“9. Mention the question number of the survey that the results described beginning with line 626 come from.” 

Answer: The cited number (n=3,117) refers to G79 (“Human origins”), and comprises the total number students who answered “true” or “false” (“active cases”), in the group of Roman Catholics in Italy and Brazil (n=2,731) (Table 1), and Non-Catholic Christians in Brazil (n=386) (Table 3), so that 2,731+386=3,117. We added more details in that paragraph to make it easier to trace back the numbers.

10. I am concerned about the somewhat strident tone in lines 655-658 regarding fundamentalist religious faith - not all fundamentalist religious people deny "the objective existence of well-founded scientific facts". I also recommend removing the word "notorious" in line 672 for the same reason. 

Answer: Both suggestions were accepted and the text was rephrased. 

“11. Which "certain conservative religions" are being referred to in lines 671-672? Be explicit.”

Answer: We introduced some changes in that paragraph, and kept the reference, which indicates a concrete example in a given sociocultural context. 

Reviewer #2

A-“The introduction discusses how knowledge and the nature of science relate to acceptance, and yet the manuscript did not seem to measure these core variables. Differences in these variables between countries could explain the acceptance patterns discussed. This was a confusing aspect of the introduction and the study”.

Answer: Our study did not aim at studying deeply the variable we were interested for: religion. We tried to throw light on the influence of religion in evolution acceptance at a certain age level, with probabilistic nationwide samples of two countries with large numbers of students who declare themselves as Roman Catholic. It is a far more modest objective that examining “how knowledge and the nature of science relate to acceptance”. However, the introduction was reformulated, trying to let our objectives clearer. 

B-“The introduction emphasizes “inconsistencies” and “differences” in prior research findings relating to acceptance and knowledge. However, the contexts of comparison (e.g., ages, countries), magnitudes of differences (e.g., effect sizes), meaning of the differences in light of the intended measures (e.g., instrument construct definitions), and quality of the studies (e.g., sampling, measurement statistics) being discussed were not carefully considered. A much more careful and rigorous review of the literature is needed. The current introduction does not carefully review studies with direct relevance to the populations of interest.”

Answer: There are several recent studies on the subject, with different methodologies and sampling methods and we were asked to present a more concise revision of previous research. As the article is not a revision, we tried to meet such a query. Thus, we introduced a consideration for the reader to consider the comparison contexts of the previously cited studies. 

C-“Line 59. The claim regarding “inconsistencies” is not supported by the literature. Studies examining different populations with different instruments and different levels of education cannot be meaningfully compared. Moreover, studies vary greatly in the quality of the measures and the rigor of the analyses. 

Answer: We removed the paragraph that compares studies with different populations, leaving the discussion on inconsistencies restricted to the study by Barnes et al. (2019) that analyzed the same group with different analysis instruments. Deleted paragraph: “"However, data are still contradictory and require further investigation. An example of inconsistencies found so far is that some studies report a strong relationship between evolution acceptance and religious beliefs, while other studies report a weak relationship between them [6]. The use of different instruments to measure evolution acceptance could be a cause of such inconsistent research results." 

D-“Line 66. The effect sizes for Barnes were not large, and the instruments utilized were intentionally designed to measure different aspects of evolution acceptance. It is NOT a flaw to intentionally conceptualize (and operationalize) a theoretical construct in different ways. Why would small differences using instruments that measure a construct in slightly different ways be “surprising”? Fahrenheit and Celcius are different scales of temperature–is one flawed because it produces a different measure? They are based on different operationalizations of theory.

Answer: Data venia, apparently the analogy does not seem to be appropriate. The mentioned piece of research found differences in the very same population using elegant random methods. If one child has two thermometers, one reads 35C and the other 101F, it is clear that the difference is small, but it does mean something important, as they point to two different realities, demanding completely different procedures. It makes no sense to choose between Fahrenheit x Celsius Scale thermometers in this case. There is no doubt that at least one instrument is not working well (if not both!). Small differences can indicate completely different realities. Our discussion focuses precisely on this point, concerning the construction of instruments. We argued that Likert scaling (see below) should not be used with factual statements, as this brings imprecise numeric measures. “I like strawberry juice” will have a far more precise measure than “strawberry is a fruit” in a Likert scale item. Biologists will not fully agree, but will not fully disagree either, with the second statement. The same would occur with pineapples, apples and pears. However, with tomato juice one could expect the opposite situation, as biologists would promptly recognize it as a fruit, but as lay costumers use to find tomatoes near potatoes in supermarkets, they may disagree with the statement, not to mention the sonority of the two words (which is not the case with Romance languages). It is not difficult to see that an instrument with several such items answered by a large and diverse population, in different countries, will bring a cloudy picture due to imprecise numeric measures. Our conclusion indicates not only that future research should not use Likert scaling with factual statements about biological evolution, but also invites reanalysis of data banks. This is precisely what we did with our own database, finding surprising results. 

E-“This introduction could be fixed by reviewing findings from high-quality studies most similar in terms of religion, age, education, etc. to the new research that will be reported.” 

Answer: We have included five paragraphs and seven references, with studies on acceptance of evolution and Christian denominations of similar age level as ours in countries such as India and Chile. The paragraphs and references are: 

 “A systematic literature review of the current state of research regarding students’ and teachers’ acceptance of evolution across Europe found that the level of acceptance of evolution in different educational settings is ambiguous. The authors argue that similar samples and a standardized assessment of the acceptance of evolution are necessary for cross-country comparisons [7]. 

 (…)

A study with secondary school students (aged 14–16) in the United Kingdom found that teaching genetics before teaching evolution had a significant impact on improving evolution understanding but did not result in a significantly increased acceptance of evolution [16]. This reflects a weak correlation between the knowledge and acceptance of evolution, which also appears to be present in undergraduate students. In Chilean students of 15–16 years old, it was found that including instruction on the nature of science in the class on evolution improved their acceptance significantly [17]. 

Kuschmierz et al. (2020) [7] identified 56 papers from the period of 2010–2020 regarding students’ and teachers’ knowledge and acceptance of evolution across 29 European countries. However, according to the authors, the research findings were hard to compare due to the application of different instruments and the assessment of different key concepts. The available database was not sufficient to obtain reasonably comparable data from European countries. In addition, the ambiguous results of the research demonstrate multiple challenges regarding measurement of evolution education, such as (a) inadequate definitions of key constructs (attitudes, acceptance, knowledge, and understanding are often not defined or inconsistently used in publications); (b) the application of diverse measurement instruments (different evolution acceptance instruments may produce different results, even when applied to the same population); (c) the multidimensionality of knowledge about evolution (most instruments focus on single evolutionary constructs, such as natural selection and related concepts).

The authors [18] then developed an instrument to measure attitudes and understanding across Europe and beyond, called the “Evolution Education Questionnaire on Acceptance and Knowledge” (EEQ). The measurement instrument was translated into several European languages, in the various Romance, Germanic, and Slavic branches, and was recently applied to 9200 first-year university students in 26 European countries [19].

This is a recent initiative, and thus, due to a lack of standardized assessment procedures in the existing literature, previous results should be used with caution when trying to compare countries, as there are several limitations, such as different age levels, and sampling procedures, etc. [18].

F-“It is not clear why the two countries are being compared if the Pew studies have shown significant differences in evolution acceptance (and prior work). A stronger and clearer rationale for the comparisons (in light of the methodological issues noted above) needs to be made so that readers can see what the authors are trying to do (which is interesting).” 

Answer: There is divergence between the two reviewers. The Pew studies were carried out with a different age level (18 years old and older), with a completely different methodology. We are emphasizing the broad sociocultural picture and the need of standardized methodological procedures, including careful item translation, aiming at producing meaningful comparisons among countries. 

G-“It was not clear how religious affiliation and religiosity were measured–were they separate? These are two well-defined constructs that are different.”

Answer: We tried to state clearly that we compared students who declared their religion, with no measure of religiosity. Possibly language editing has solved this problem. 

H-“The methods discuss measuring evolution acceptance, but later on the discussion mentions ‘understanding’ in several places. It was not clear if there is a knowledge measure and an acceptance measure. Or are the authors conflating acceptance and understanding? This was confusing. (e.g., “Difference of geological time understanding followed the same pattern, with a very high score”)”. 

Answer: We rephrased the parts where we present the only item that relates to geological time. It is not included in the final figure, which tried to show the influence of religion on evolution acceptance. 

I-“Line 250. It is important to consider that Likert scales by themselves are not instruments or measures. Measures are derived from the scale, and such scales pose many problems (e.g., non-ratio scaling) that can distort inferences. The way that these issues are discussed is problematic and should be revised.”

Answer: We deleted the phrase “The Likert scale is considered a good instrument to estimate the degree of agreement (or disagreement) about statements expressing opinions.”, and inserted a more precise description of our main criticism of previous instruments: 

 “Likert scaling is used in instruments measuring attitudes, beliefs, and opinions about statements, allowing expression of moderation of opinion from agreement to disagreement. The key point is finding ways of calibrating how strongly or mildly a statement should be worded. These scales are therefore suitable for issues related to open constructs and less for scientific claims based on presumed factuality. An item likely to produce extreme responses, either full agreement or strong disagreement, would do a poor job of discriminating across the full spectrum of respondents [57]. Therefore, Likert scales may not be a good choice with statements expressing well-known scientific facts. In addition, in the case of theories of evolution, factuality is supported by the school context in which the questionnaires were filled out. The battery of items on evolution was proposed with dichotomous questions given that each of them can have minimal variability”. 

A previous answer (#D) brought additional elements for this discussion, but may not be relevant in the context of the manuscript, considering writing style and content. 

K-“ Line 288. The introduction emphasizes that different instruments that measure constructs in different ways produce different results, and so it was surprising to read that a new instrument that has not been widely used was employed in this study. How, then, can the results be compared to prior work? It seems that the problems emphasized in the introduction are ignored and the authors continue the tradition of comparing patterns using different tools. How will this move research forward? The methodological approach was puzzling. Regardless, evidence in support of instrument quality needs to be expanded as it is inappropriate to interpret measures unless there is a robust body of evidence supporting them (see Mead et al. Evolution Education Outreach on instrument quality in evolution education). Please report comparable evidence types for the instrument.”

Answer: The issue of comparison was already addressed (see above). In addition, we state clearly that our instrument was based on previous ones, and not exactly a brand new one, although we did not use Likert scale. 

L-“ The sampling methods should include information on missing data and whether it was missing at random (e.g., did students of particular groups have disproportionate missing data?). The participation rate should be mentioned in the main text.”

Answer: This criticism was, in fact, the most profound and challenging one. Our database was searched on five occasions, including the doctoral thesis of a specialist in statistics, who studied precisely the existing clusters. However, as it is a relatively large database, at least in the context of the ROSE Project, the groupings always present overlaps that do not allow to figure out so clearly the image revealed by the MCA study, focusing on the same religion in the two countries. It led us to perform the easy-to-understand numerical calculations that resulted in the polygon image. Although we had no evidence that the missing cases could be biased, it was necessary to carry out a specific study. Thus, a new analysis of the database was performed and, now, we can categorically state that there are no elements that indicate any bias in the missing cases of the MCA. A new set of tables was added to Supplemental Materials (S2 File). A paragraph with the conclusion of the analysis was included and reads: 

 “The detailed analysis of the occurrence of missing values allows us to conclude that there is no indication that the MCA results have any significant bias when comparing the responses of valid active cases with those with missing values.” 

We included the participation rate in the main text, and kept the summary and matrix of the MCA as supplementary materials. We are very grateful for the observation, which disturbed our sleep for a few days, but which, in the end, made our conclusions more solid. 

M-“The data have clear nesting and HLM (hierarchical linear models) would allow for the analyses of patterns within schools, regions, and countries. The current methodological approach is very unusual and so a rationale is needed for why MCA was used.”

Answer: As mentioned in the previous answer, usual tests were performed on five doctoral theses, which resulted in several articles. However, it was not clear that the differences between Roman Catholic students from the two countries could be greater than the differences between them and Non-Catholic Christian denominations in the same country. 

N-“Why were the data on nationality and religion used to code responses when it would be possible to use predictor variables contained within the dataset? In other words, the analyses ignore the richness of the dataset and predispose the results to the coding scheme. This is problematic and poorly justified. Within- and between- group variation can be modeled along with interactions.”

Answer: As mentioned in the previous answers, the richness of the data was studied in five doctoral theses, which resulted in several articles. One of them (Ocampo & Tolentino Neto, 2020) found different typologies of students, which were determined based on the hierarchical clustering method. However, the isolation of country and religion from the acceptance of evolution was yet to be done. Possibly, the results published by research groups from different countries, after our data collection, and the growth of conservative and anti-evolutionist influence, mainly in Brazil, have been decisive to emphasize the need for this particular study. 

O-“The discussion section should emphasize why these results matter. Let’s say that the results are well supported by evidence (which is by no means clear). How does this change how biologists should approach the challenge of evolution education? What are the implications for other countries? How should sociocultural factors be measured and included in models of acceptance? The discussion section should also include prior evolution education research in non-religious contexts such as China and other international studies (please review Donnelly and Deniz Springer and other studies).”

Answer: We included new references, but could not include the mentioned article (probably Deniz, Donnelly and Yilmaz, 2008), where authors discuss other components in someone’s conceptual ecology for biological evolution. We believe we pointed out several implications of our results, including revising databanks and methodological aspects regarding items construction exploring factual scientific statements about biological evolution. 

Cited Bibliography (not included in the article):

Deniz, H., Donnelly, L.A. and Yilmaz, I. Exploring the factors related to acceptance of evolutionary theory among Turkish preservice biology teachers: Toward a more informative conceptual ecology for biological evolution. Journal of Research in Science Teaching 45 (4): 420-443, 2008. 

Dixon, Thomas. Science and Religion: A very short introduction. Oxford: Oxford University Press, 2008.

Haught, John F. Science and Religion: from conflict to conversation. Mahwah (NJ): Paulist Press, 1995.

Ocampo, D; Tolentino-Neto. As diferentes tipologias que descrevem o interesse dos jovens brasileiors pelas ciências [The different typologies describing the interest of Brazilian youth in Science]. Amazônia/Revista de Educação em Ciência e Matemática 16(37): 164-176, 2020. [available from: https://www.periodicos.ufpa.br/index.php/revistaamazonia/article/view/8660/6696>.].

Smith, Christian. Religion: What it is, How it works, and Why it Matters. Princeton (NJ): Princeton University Press, 2017.

Smith, Christian, Christian America? What evangelicals really want. Berkeley: California University Press, 2000.

---

## [Editor Report · Decision Letter 1]

18 Aug 2022

Acceptance of Evolution by High School Students: is religion the key factor?

PONE-D-22-03227R1

Dear Dr. Bizzo,

We’re pleased to inform you that your manuscript has been judged scientifically suitable for publication and will be formally accepted for publication once it meets all outstanding technical requirements.

Kind regards,

Norman Johnson

Academic Editor

PLOS ONE

Additional Editor Comments (optional):

The author addressed concerns of reviewers to my satisfaction.
---

## [Editor Report · Acceptance letter]

23 Aug 2022

PONE-D-22-03227R1 

Acceptance of Evolution by High School Students: is religion the key factor? 

Dear Dr. Bizzo:

I'm pleased to inform you that your manuscript has been deemed suitable for publication in PLOS ONE. Congratulations! Your manuscript is now with our production department. 

Kind regards, 

on behalf of

Dr. Norman Johnson 

Academic Editor

PLOS ONE